# Social Media Big Data Mining and Spatio-Temporal Analysis on Public Emotions for Disaster Mitigation

**Tengfei Yang** [1,2], **Jibo Xie** [1,*], **Guoqing Li** [1], **Naixia Mou** [3], **Zhenyu Li** [3], **Chuanzhao Tian** [1,2] **and Jing Zhao** [1,2]

1    Institute of Remote Sensing and Digital Earth, Chinese Academy of Sciences, Beijing 100094, China; yangtf@radi.ac.cn (T.Y.); ligq@radi.ac.cn (G.L.); tiancz@radi.ac.cn (C.T.); zhaojing01@radi.ac.cn (J.Z.)
2    University of Chinese Academy of Sciences, Beijing 100094, China
3    College of Geomatics, Shandong University of Science and Technology, Qingdao 266590, China; mounx@lreis.ac.cn (N.M.); lizy1@radi.ac.cn (Z.L.)
*    Correspondence: xiejb@radi.ac.cn

**Abstract:** Social media contains a lot of geographic information and has been one of the more important data sources for hazard mitigation. Compared with the traditional means of disaster-related geographic information collection methods, social media has the characteristics of real-time information provision and low cost. Due to the development of big data mining technologies, it is now easier to extract useful disaster-related geographic information from social media big data. Additionally, many researchers have used related technology to study social media for disaster mitigation. However, few researchers have considered the extraction of public emotions (especially fine-grained emotions) as an attribute of disaster-related geographic information to aid in disaster mitigation. Combined with the powerful spatio-temporal analysis capabilities of geographical information systems (GISs), the public emotional information contained in social media could help us to understand disasters in more detail than can be obtained from traditional methods. However, the social media data is quite complex and fragmented, both in terms of format and semantics, especially for Chinese social media. Therefore, a more efficient algorithm is needed. In this paper, we consider the earthquake that happened in Ya'an, China in 2013 as a case study and introduce the deep learning method to extract fine-grained public emotional information from Chinese social media big data to assist in disaster analysis. By combining this with other geographic information data (such population density distribution data, POI (point of interest) data, etc.), we can further assist in the assessment of affected populations, explore emotional movement law, and optimize disaster mitigation strategies.

**Keywords:** social media; big data; fine-grained emotion classification; spatio-temporal analysis; hazard mitigation

## 1. Introduction

With the popularity of mobile devices and the development of the network infrastructure, social media has quickly integrated into people's lives. People can easily share what they see and hear, and even what they feel and think with social media. They are like "mobile sensors" [1] to collect information around them constantly. This provides a new way to acquire disaster-related data. Compared with traditional disaster information collection methods, social media has the characteristics of real-time information provision and low cost. Furthermore, these data contain a lot of geographic information (such as location, time, and other attribute information), which is very important for disaster mitigation. Therefore, many researchers have noticed the importance of social media in

disaster mitigation. They have studied disasters from the perspectives of event extraction [2,3], user trajectory rules [4] and data fusion [5], etc., and achieved good results. However, few researchers have considered the public emotional information contained in social media (especially fine-grained emotions) as an attribute of disaster-related geographic information to aid in disaster mitigation. When disasters occur, public emotions often express the public's attitude towards disaster, needs during disaster, and feedback on disaster relief, etc. These are very helpful to understand the progress of the disaster quickly and effectively improve the efficiency of rescue. However, there is still a lack of an effective framework to quickly collect, process, and use this emotional information. There are three problems involved: (1) How can the fine-grained public emotional categories be divided during the disaster? (2) Social media has a huge user base. We take Sina micro-blog, a Chinese social media, as an example. According to statistics, as of Q3 2018, Chinese social media platform Sina micro-blog had over 431 million active monthly users [6]. When disasters occur, this will generate a lot of disaster-related data. As such, how can the fine-grained emotional information contained in these data be extracted more accurately? (3) When these fine-grained emotions are extracted, how can they be regarded as an attribute of disaster-related geographic information to assist disaster mitigation? In this paper, we used a Sina micro-blog and took an earthquake disaster as an example to describe how the framework we built extracted fine-grained public emotions and used them to serve disaster mitigation.

Unlike most emotion analysis studies (they usually divide emotions into three categories: positive, neutral, and negative), we divide the public emotions during the disaster into more dimensions, because the use of multiple dimensions of emotion in the disaster context can allow more details of the disaster to be described. Additionally, studies have illustrated the importance of multidimensional emotional information in disasters. Ekman, et al [7] showed the differences between anger, disgust, fear, and sadness in terms of antecedent events and likely behavioral responses. Oliver Gruebner et al [8] analyzed how to apply multiple dimensions of negative emotion (including anger, fear, sadness, surprise, confusion, disgust) to survey disaster mental health. Existing psychological studies [9–11] also mention the fine-grained division of emotions in a disaster. Therefore, based on these previous studies and the corpus used in this paper, we subdivide the negative emotions into anger, anxiousness, fear, and sadness.

The commonly used methods for emotion classification include rule-based algorithms and traditional machine learning models [12]. Rule-based algorithms mainly uses given emotional lexicons and corresponding grammatical rules to calculate the emotional intensity of the text [13,14]. This method relies on a large number of manual operations, such as manual development of search rules and a large-scale emotional lexicon [15], which determines the accuracy of the method. Additionally, this method is weak in dealing with stop words and new words. It is also hard to add some slang and Internet buzzwords to the emotional lexicon in time, such as "喜大普奔" (great satisfaction), "狂顶" (very supportive), etc., which often appear in social media. Traditional machine learning models, such as naive Bayes [16], maximum entropy, and support vector machine [17] do not rely on emotional lexicons or search rules. They only need to manually annotate the training set. However, the traditional machine learning method is based on the bag-of-words model, which ignores the semantic relations in text. In other words, it does not consider the order of words in a sentence, which can easily cause misclassification of emotions. For example, the sentences "Although the earthquake is terrible, we are safe and sound" and "Although we are safe and sound, the earthquake is terrible" contain the same words, they express different emotions. Moreover, for traditional machine learning models, the input is the feature words extracted from the text after segmentation. The definition of feature words has a significant impact on the model's efficiency [15]. We selected the deep learning method to extract public emotion from social media. Compared with the rule-based method, deep learning does not depend on any emotional lexicons. Therefore, it is not affected by new and unknown words. Unlike traditional machine learning, deep learning uses word vector models to replace the bag-of-words model, which can make good use of semantic information in sentences. Much research has indicated

that the performance of deep learning [18,19] in natural language processing (NLP) tasks is better than of traditional machine learning.

Furthermore, we used extracted fine-grained public emotions and combined them with traditional geographic information data (population density distribution data, point of interest (POI) data, etc.), and the powerful spatial analysis functions of a GIS (geographic information system) to assist disaster relief. Combining public emotional information could produce the following benefits: (1) It could improve the accuracy and efficiency of disaster assessment. For example, with the help of the powerful spatial analysis functions of a GIS, traditional geographic information data (such as population distribution, traffic distribution, etc.), and emotional distribution data can be combined to assess the affected population in real-time. People who express negative emotions are generally considered to be more affected by disasters. (2) It could help to reduce disaster-related losses. For example, disasters, especially sudden disasters (such as earthquakes, volcanic eruptions, etc.), can easily cause disaster-related mental health problems, such as post-traumatic stress disorder (PTSD) and depression [20–22]. Traditional monitoring has difficulty obtaining information about emotions of the public in the disaster area (despite the existence of a questionnaire, its real-time performance is poor). If information about the public's emotions and corresponding spatio-temporal distribution is known, the disaster reduction department can take corresponding psychological rescue measures to reduce the occurrence of disaster-related mental health problems. In addition, extreme disasters have the characteristics of inevitability and unpredictability [23]. People will express different emotions at different stages [24] and have different responses to try to overcome them [9]. For example, anxious people are more sensitive to the negative side of event-related information and can be easily influenced by rumors [25]. Therefore, through understanding the distribution of anxious people, we can release the correct disaster information at appropriate times to prevent rumors being intrusive to anxious people. (3) Learning more about the causes of emotion could help us to optimize emergency decisions. By using different emotion categories, we can explore different emotional causes, such as why angry emotions are predominant in a certain area and more anxious emotions are predominant in another area, and why the emotion categories change in some places over time. By understanding the causes of emotion, the disaster reduction departments can carry out targeted countermeasures. In the spatio-temporal analysis of public emotion information, the framework in this paper includes assessing the affected population in real-time, exploring an emotional movement law, and monitoring the causes of emotional change.

The structure of this paper is as follows: Section 2 describes the data acquisition, parsing, processing, and emotion classification method used in this paper. Section 3 presents the role of public emotion in assisting disaster reduction with a case study. Section 4 shows the evaluation of the experimental indicators. Section 5 concludes the paper.

## 2. Framework to Analyze Public Emotion from Social Media Big Data

The framework to analyze the role of public emotions in disaster mitigation proposed in this paper includes five major phases: data acquisition and processing, the construction of a word vector list, model training, emotion classification, and spatio-temporal analysis of public emotions (as shown in Figure 1).

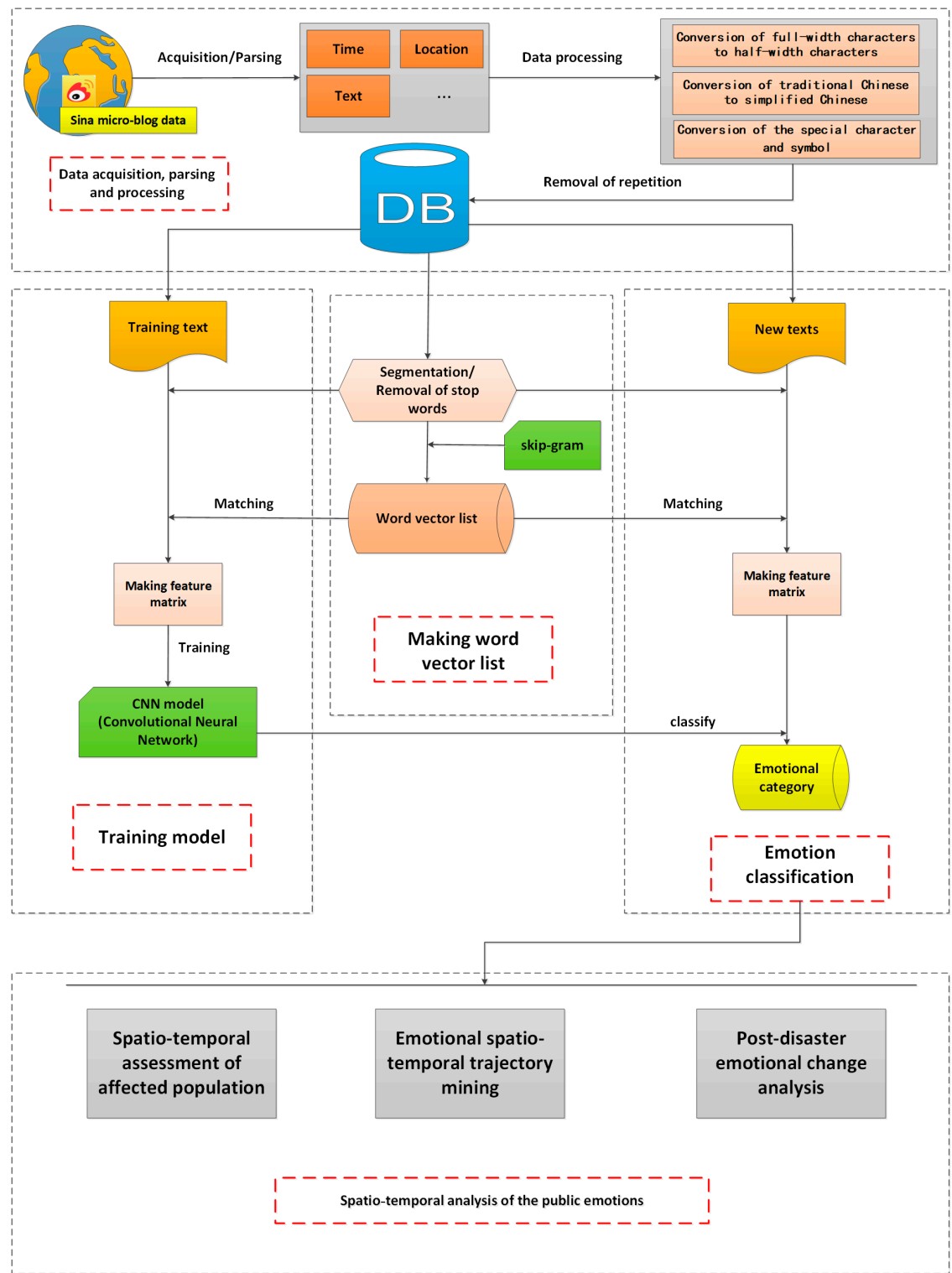

**Figure 1.** Framework of the automatic emotion classification and disaster analysis.

### 2.1. Social Media Data Acquisition and Parsing

We used an earthquake that happened in Ya'an, Sichuan, China, at 08:02 h on April 20, 2013, as the case study. According to the report by the China Seismograph Network (http://news.ceic. ac.cn/CC20130420080246.html), the magnitude of this earthquake was 7.0 and its focal depth was 13 kilometers. The epicenter of this earthquake was located at 30.30° N, 103.00° E, which caused about 1.52 million people to be affected in an area of 12,500 square kilometers.

In this paper, social media data was acquired from the Sina micro-blog from the region surrounding the epicenter with a radius of 200 km, which was severely damaged by the earthquake. The affected cities included Ya'an, Meishan, Ganzi, Leshan, Ziyang, Deyang, Chengdu, Aba, Zigong, Mianyang, and Neijiang, as shown in Figure 2. The time span of social media data was from April 20 until April 26, 2017. Social media platforms usually provide an interface or API (Application Programming Interface) that allows developers to retrieve social media data. However, the retrieval of data in this way has great limitations; for example, you cannot set the time-span and topics, etc. Therefore, in this paper, we used Sina micro-blog's advanced search capability to get data by using time-span, city names, and event-related key words.

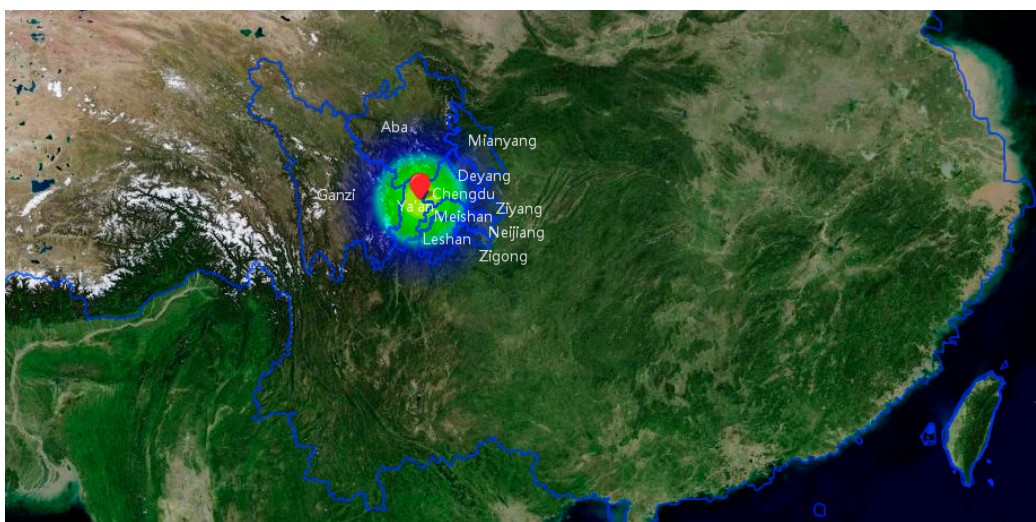

**Figure 2.** The study area of the 2013 Ya'an earthquake that was used in this paper.

The data format was initially hypertext markup language (HTML). We parsed the data into a structured data format including fields such as "time," "location," "text," etc. Among them, location was represented by the address and the accuracy of them were different. We take Chengdu as an example. Some addresses were described in more detail, such as "East Gate of Sichuan University," "Sishengci North Street," etc. Some addresses were roughly described, such as "Funan New District." There were also some texts that did not have address information. The reason for this is that people have different usage habits (some people do not want to share their location information). We used the API provided by Baidu (http://lbsyun.baidu.com/index.php?title=webapi/guide/webservice-geocoding) to convert these addresses to latitude and longitude. Among them, for line data, such as "Sishengci North Street," we took its midpoint coordinates to represent it. For surface data, such as "Wangjiang Campus of Sichuan University" and even "Funan New District," we extracted the central point coordinate to represent them respectively. We did not assign coordinates to those texts that did not have address information, including those with rough addresses. They were just labeled "Chengdu."

### 2.2. Social Media Data Processing

In the subsequence processing steps, we mainly dealt with the text data. The main text processing steps included the conversion of full-width characters to half-width characters and from traditional Chinese to simplified Chinese, as well as recognition of the special characters and symbols. The aim of the first two steps was mainly to improve the computational efficiency of the model. The third step aimed to recognize special characters and symbols, such as "(>_<)", "☺", which are deleted and ignored by many common natural language processing (NLP) tools. However, for an emotional analysis, these special characters and symbols have emotional meaning, for example, "(>_<)" and "(>_<)>" can express troubled emotions. Therefore, in this paper, we interpreted them into text that could be processed by NLP. Some special characters and symbols could be translated into text by the

micro-blog platform. For example, 😭 could be translated into "tear" (泪). However, others that could not be decoded by the micro-blog platform, such as (>_<) and 💔, were interpreted according to the web's "list of emoticons," which includes the emotional implications of all kinds of emoticons through a large amount of published literature. For example, (>_<) can be translated into "troubled" (焦虑) and 💔 can be translated into "sad" (伤心).

Finally, after eliminating duplications, there were 39341 data records stored in our database.

### 2.3. Constructing the Word Vector List

In this paper, we first converted each word from the previously processed texts into a multidimensional vector. This process included two phases: word segmentation and the removal of stop words, and the construction of a word vector list.

### 2.3.1. Word Segmentation and the Removal of Stop Words

Unlike the English language, there is no space separation between Chinese words. Therefore, we needed to segment Chinese text to get separate words. Additionally, the Chinese micro-blog is more colloquial, which brings great challenges to word segmentation. We compared many different Chinese word segmentation tools, such as "Stanford NLP", "ANSJ", "NLPIR (Natural Language Processing & Information Retrieval Sharing Platform)", and so on. We found that "NLPIR" had the best performance in terms of the accuracy and speed of word segmentation.

There are many meaningless words in text after word segmentation; these are called stop words, such as "在 (on)," "是 (is)," "一会 (a moment)," and so on. These words could affect the accuracy of the model and therefore should be removed. In this paper, we used the vocabulary of stop words developed by the Harbin Institute of Technology—Social Computing and Information Retrieval Research Center to remove stop words. As the focus of this paper was on the emotional analysis, we optimized the vocabulary of stop words by removing sentimental words, such as "愤然 (indignant)," "幸亏 (luckily)," "嘻 (hey)", etc.

### 2.3.2. Construction of the Word Vector List

The input used for the emotion classification model was a word vector matrix. We needed to convert each word in the micro-blog text into a multidimensional vector, and then convert the whole sentence into a word vector matrix. In this paper, we converted all previously processed texts into a word vector list. The training text and new text to be categorized were transformed into a word vector matrix by matching them with the word vector list. The method we used for this was word2vec [26], which projected every word in every sentence to a specified dimensional vector space.

There are two commonly used models in word2vec, which are skip-gram [27,28] and CBOW (Continuous Bag-of-Words) [27,28]. A large number of experiments have been done to compare these two models in terms of performance and accuracy [29] and the results show that the semantic accuracy rate of the skip-gram model is better than that of the CBOW model. Therefore, we used the Skip-gram model in our experiment to construct the text feature vector.

The skip-gram model can determine correlations between words for corpus training. These correlations are represented by the multidimensional feature vectors of each word. Additionally, these multidimensional feature vectors are calculated by taking full consideration of the context of semantic information. From the below formula, given a current word, $w_i$, this model tries to find words which have a contextual semantic relationship with the current word. The target of this model is to maximize the objective function, $G$:

$$G = \sum_{w_i \in C} \log P(Context(w_i)|w_i). \tag{1}$$

In this formula, $w_i$ represents the current word and $C$ represents the context window. $P(Context(w_i)|w_i)$ represents the probability of the context information in the current word.

When the training converges, words with similar semantic meanings are closer in the specified dimensional vector space. We exported the text feature vector of each word in the training corpus to generate word embedding. The structure of text feature vectors is shown as Figure 3.

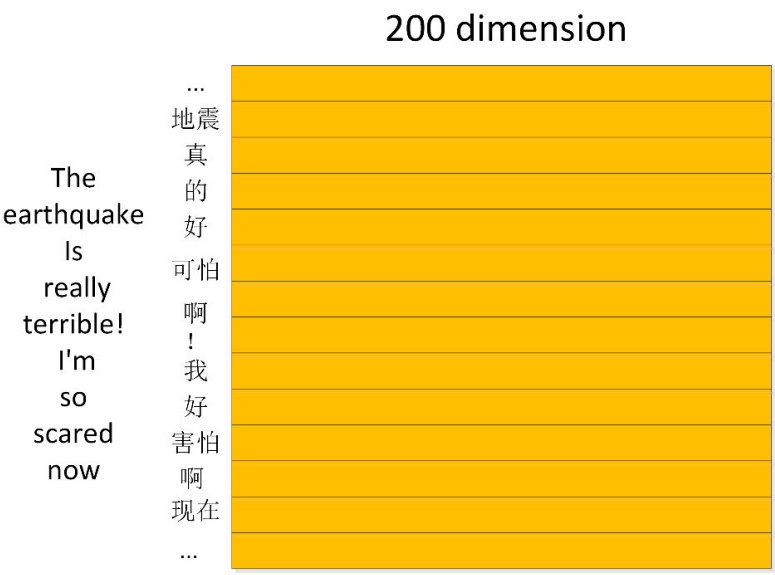

**Figure 3.** Structure of the text feature vector.

### 2.4. Model Training

The deep leaning model selected in this paper was the convolutional neural network. We read much related literature and found that different deep learning methods can be selected for emotion classification, such as the convolutional neural network (CNN), recurrent neural network (RNN), hierarchy attention network (HAN), etc. These models [30–32] all have their own characteristics and usage scenarios. According to the literature [33], CNN performs emotion classification well, especially in shorter sentences. RNN performs document-level emotion classification well [34]. A previous study [35] presented the performances of the CNN, RNN, and HAN in emotion classification. The results showed that when the training corpus is large enough, HAN has the highest accuracy, but CNN performs the best when the training corpus is not very large. The annotation of a large training corpus requires a lot of manpower and time. Additionally, it takes longer to train the HAN and RNN models than the CNN model. In this paper, the training corpus used was micro-blog texts, which are mainly short texts. Additionally, the amount of manually tagged training corpus data was more suitable for the CNN model. Therefore, we selected CNN as the method to extract the public emotion contained in social media. The training process of the model is shown below.

#### 2.4.1. Word Segmentation, Removal of Stop Words, and Construction of the Feature Matrix

First, we segmented the training texts to obtain separate words. Then, we used a vocabulary of stop words to remove the meaningless words contained in those separate words. Finally, the remaining words were converted into word vectors through matching with a previously generated word vector list. Ultimately, every sentence was transformed into a feature matrix.

#### 2.4.2. Training Convolutional Neural Network Model

The convolutional neural network (CNN) is a variant of the neural network. It was first successfully used for the recognition of images and videos. Later, some researchers introduced it into the field of natural language processing [36] and found that it had a good effect. The CNN

model used in this paper consisted of an input layer, a convolutional layer, a pooling layer, a fully connected layer, and classification. The structure of the CNN is shown in Figure 4.

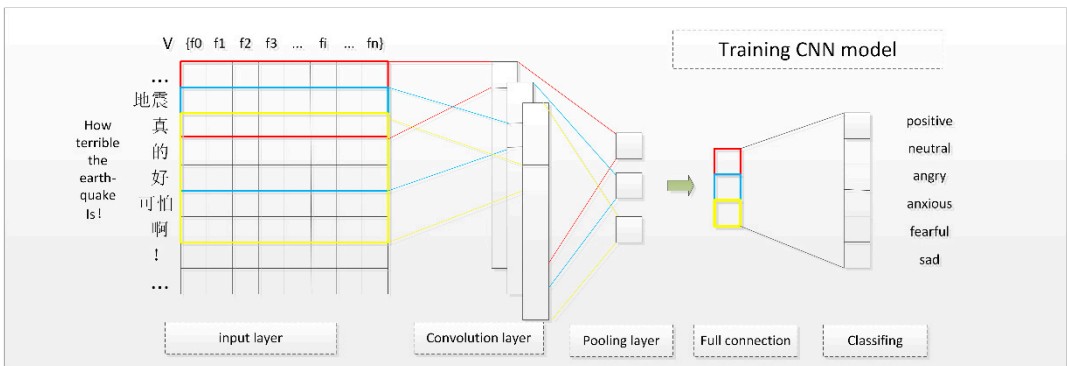

**Figure 4.** Structure of the convolutional neural network (CNN) model used in the paper.

In the course of the training process of the CNN model, the neurons in it are usually set to three dimensions: depth, width, and height. The size of each layer is depth × width × height [37]. For example, if there is a sentence with 140 words, and each word is set to be 200 dimensions, the size of the input layer is 1 × 140 × 200.

Next, we introduce the layers of the convolution neural network.

*Input layer*: The input layer of CNN is a matrix that consists of text feature vectors. This matrix is calculated using the skip-gram model, which was described in Section 2.2. The rows and columns (dimension) in this matrix were set before we put the matrix into the neural network model. Taking the Sina micro-blog text as an example, the number of characters in each sentence was less than 140. Therefore, we set the rows in the matrix to 140. If the number of words in a sentence was less than 140 characters, we used the empty character "space" to supplement the missing characters. Therefore, every sentence is expressed as follows:

$$S_{1:140} = S_1 \oplus S_2 \oplus S_3 \ldots \oplus S_{140.} \tag{2}$$

In this formula, S represents a character or "pad," and $\oplus$ is the concatenation operator.

*Convolutional layer*: The convolution layer is mainly used to extract features. It abstracts some fragmented elements into features which can be used to distinguish different categories. By convolution, many low-level features can be abstracted to higher level features. For example, the single word "打" or "call" has no emotional meaning. However, the higher-level feature "打call (praise)" can express an emotional attribute. The emotional attributes of these words can be acquired by the model through a large number of training corpus.

Given a matrix $u$ that is from the input layer for convolution operation, the formula is as follows:

$$c_j = f\left(u * k_j + b_j\right). \tag{3}$$

For the matrix $\mathbf{u} \in \mathbb{R}^{D \times L}$, $D$ represents the embedding dimensionality, and $L$ represents the sentence length. The parameter $\mathbf{k} \in \mathbb{R}^{D \times s}$ represents the $j$-th convolutional kernel, which is applied to a window of $s$ words. The parameter $b_j \in \mathbb{R}$ represents a bias term. $f\left(u * k_j + b_j\right)$ is a non-linear activation function.

*Pooling layer*: After the convolution operation, we can use the output features to directly classify emotions. However, in doing so, we will not only face the challenge of computational complexity, but also the problem of over-fitting, which will affect the classification accuracy. The pooling operation can solve these problems well. In addition, the pooling operation can also serve as a feature selector that can help to identify the most important features to improve the classification performance.

There are two methods that can be selected, namely max pooling and average pooling. We achieved better results with the max pooling method. This method selects global semantic features and attempts to capture the most important feature with the highest value for each feature map [37]. The output from the convolution operation $c_j$ is used as the input of the pooling operation. The formula is as follows:

$$p_j = pooling\left(c_j\right) + b_j. \tag{4}$$

*Fully Connected Layer*: The neurons in this layer have full connections with all neurons in the previous layer. Meanwhile, the value of the full connected layer can be calculated through the neurons in the previous layer. In the calculation process, the dropout regularization method is usually used to avoid over-fitting.

*Classifying*: We can obtain the emotional labels of the original text through the softmax function. In other words, these calculated results represent the probability distribution of the emotional labels.

Based on the training corpus, we can identify the best parameters for the CNN model. Then, this trained model can be used to calculate the emotional categories of new texts.

## 2.5. Emotion Classification

We used the trained CNN model to analyze new texts. The emotions contained in these texts were divided into six categories: positive, neutral, angry, anxious, fearful, and sad. Among them, the positive emotion mainly included the public's satisfaction with disaster relief, the public's wishes for the disaster area, and the joy of surviving. The neutral emotion mainly included objective descriptions of the disaster. In the process of classification, new texts were first processed using word segmentation and the removal of stop words. Then, the previously trained word vector list was used to translate each word into a word vector. Furthermore, each new text was transformed into a word vector matrix. Finally, the word vector matrix was input into the trained CNN model. Through the calculation of the model, each new text was labeled into the different emotional categories. We classified all 39,344 pieces of texts into the six emotion categories based on this classification process.

## 2.6. Spatio-Temporal Analysis of the Public Emotions

The framework in this paper aims to assist with disaster mitigation by using the public emotional information contained in social media. In the process, emotional information was regarded as an attribute of geographic information. The powerful spatial analysis capability of a GIS was used to combine the emotional information with other geographic data to dig out more useful knowledge. For example, the population density distribution data can be added to carry out a spatio-temporal assessment of the affected population. The POI data (such as a sanctuary) can be considered to explore the spatio-temporal trajectory law of people in sudden disasters. In addition, emotional information can also help disaster reduction departments to screen out urgent public demands from a vast amount of information. The public demands that contain emotional information are also effective feedback for disaster reduction work. They can help to optimize decision-making to improve rescue efficiency.

## 3. Spatio-Temporal Analysis of Public Emotional Information

### 3.1. Spatio-Temporal Assessment of the Affected Population

It is very important to know the distribution of the affected population at the time when an earthquake happens. This can help to ensure an effective assessment of the disaster situation and rational deployment of rescue resources. In this section, we combined the population density distribution data related to the study area with the spatio-temporal information contained in social media to assist analysis. Among them, the population density distribution data was taken from the GHSL (Global Human Settlement Layer) (http://cidportal.jrc.ec.europa.eu/ftp/jrc-opendata/GHSL/GHS_POP_GPW4_GLOBE_R2015A/). The introduction of public emotional information can improve the accuracy of the assessment. It is generally believed that negative emotions indicate that

the earthquake has a greater impact on the people. Further, according to the rules of time period division from the rescue department, we set up six time periods, which were: 0–4 hours, 4–12 hours, 12–24 hours, 24–48 hours, 48–72 hours, and 0–72 hours after the earthquake. Then, we used the overlay analysis of GIS software to process these data in each time period. The results of the analysis of the related data are shown in Figure 5.

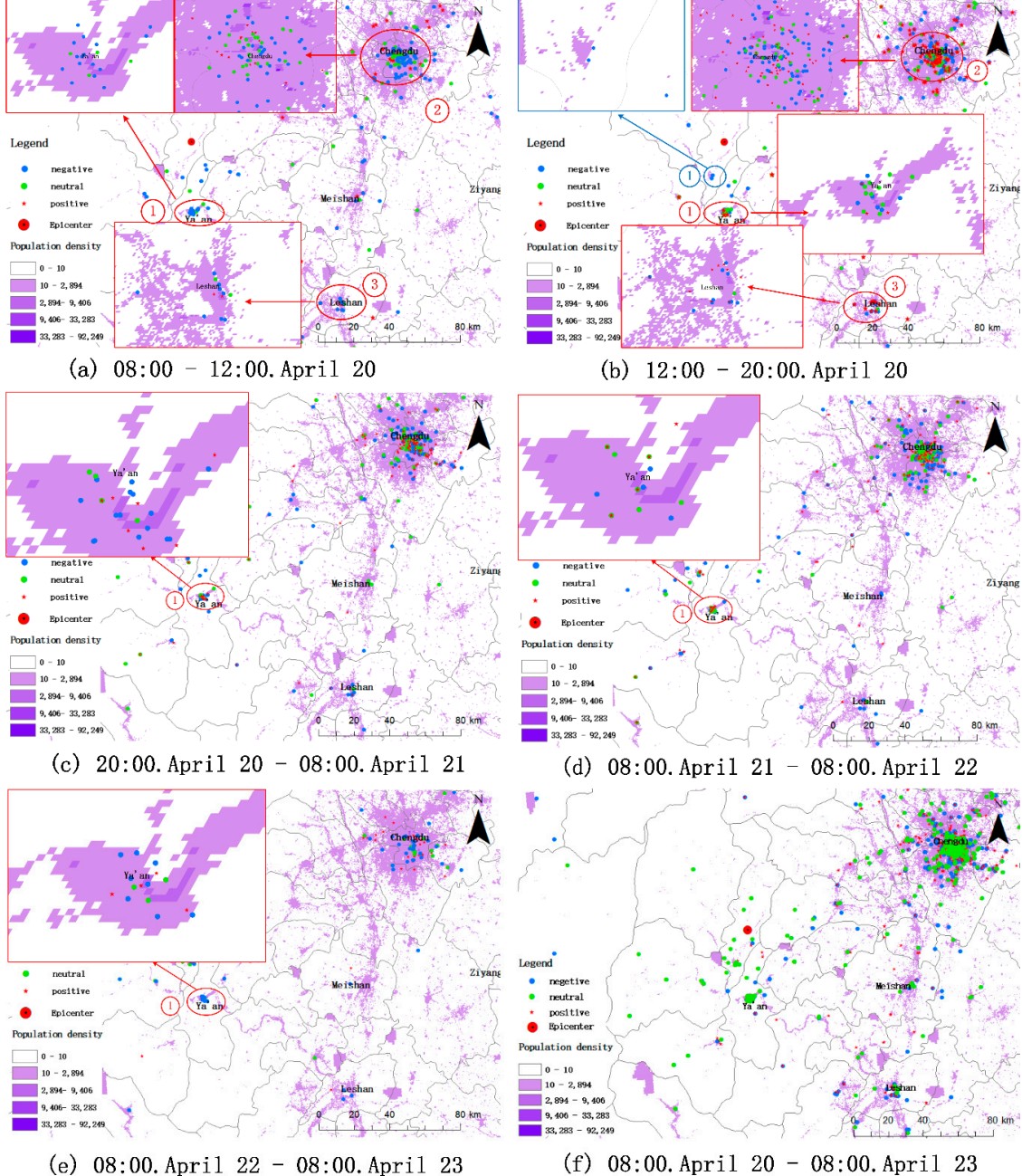

**Figure 5.** Emotional distribution characteristics of the affected population. The figure (**a**), (**b**), (**c**), (**d**) and (**e**) describe the distribution of emotions in different time periods within 72 hours after the disaster. The figure (**f**) shows the distribution of emotions over 72 hours. Among them, each of red circle 1, red circle 2, and red circle 3 in the figures represent the same area. The blue circle 1 in (**b**) shows that compared with (**a**), new negative emotions emerged in same area.

We know that: (1) The microblog data volume was larger in places with a high population density after the earthquake and negative emotions predominated. (2) Within four hours after the earthquake,

as shown in Figure 5a, there were almost no positive emotions in the areas near to the epicenter. Areas far away from the epicenter, such as Leshan (red circle 3) and Chengdu (red circle 2), had fewer positive emotions. (3) From 4 hours to 12 hours after the earthquake, as shown in Figure 5b, compared with the previous distribution of emotional information, some new negative emotions emerged near the epicenter, such as the blue circle 1. This indicates that as time went on, some new disaster damage might have taken place in this region. We checked the corresponding text and found that these new emotion points were mostly anxiety. The reason people expressed anxiety was because the "Fan Min Road" was blocked by boulders, and people were worried that the rescue vehicles that did not know this information which would be delayed due to the incident. The emotions in other areas of this figure had also changed. For example, compared with the red circle 2 and red circle 3 in Figure 5a, the emotions in these areas of Figure 5b increased significantly. This indicates that the public's attention to earthquakes continued to increase in this time period. (4) We selected an area with a high population density near the epicenter to analyze how emotions changed over time in detail. The selected area was located in Yucheng District in Ya'an and it was marked with red circle 1 in Figure 5a–e. Figure 6 shows the changes in data volume in emotion categories in this area for different periods of time. We found that the positive emotions began to appear in the second period and then continued to increase. The reason was that as the rescue operation unfolded, people's gratitude for the rescuers increased. The number of negative emotions increased the most in the third period and then gradually reduced. Although this time period lasted only 12 hours, the number of negative emotions was the highest of all periods. Because this time period was the first night after the earthquake, most people were in urgent need of relief supplies, such as tents, clothes, etc. Therefore, anxiety was dominant. The number of neutral emotions expressed did not change very much. They mainly described the progress of earthquakes. (5) Figure 5f depicts the overall situation 72 hours after the earthquake. We found that the population density in Ya'an was not high, and the distribution of population was not uniform. However, the number of emotions expressed in this city was large and the distribution of them was fairly uniform, especially regarding negative emotions. This shows that the impact of the earthquake on Ya'an was the most serious. In addition, Chengdu is the capital of Sichuan Province and has the largest population density. During this earthquake, many negative emotions were expressed in this city. Although the distribution of these emotions was not uniform, more attention should be placed here to avoid unexpected accidents, such as people being hurt by rumors due to anxious emotions. The same approach can also be applied to other affected cities.

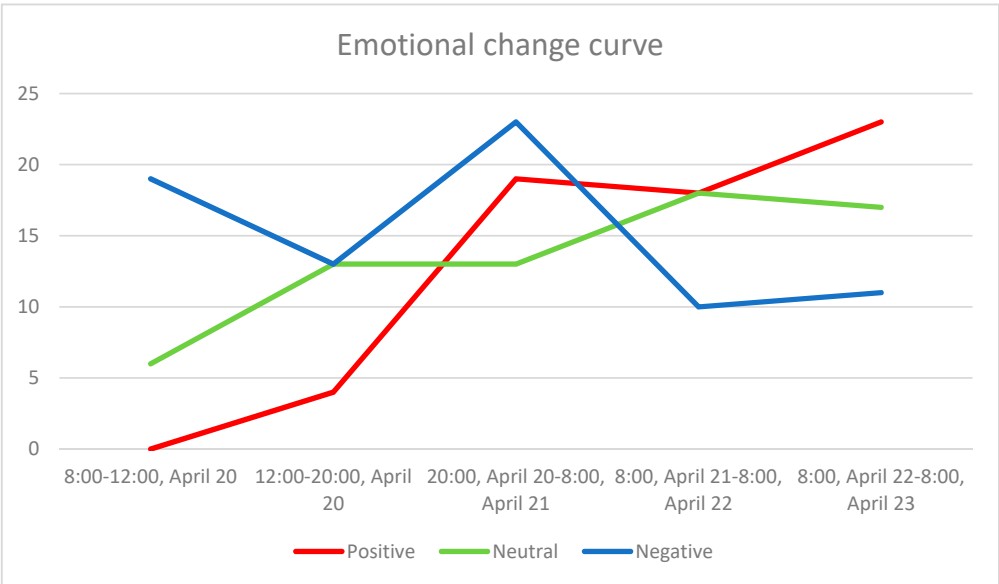

**Figure 6.** Changes in different emotion categories in data volume for different periods of time.

In this section, we conducted an overlay analysis containing public emotional information and population density distribution to assess the affected population. Spatio-temporal distribution characteristics of public emotional information can improve the accuracy of assessment and provides us with more valuable information. Although this emotional information was unevenly distributed, and even some areas with high population density had only a small amount of negative emotions, such as the area in the blue circle 1 in Figure 5b, we should still pay attention to them because emotions expressed by users of social media may also reflect emotions from the neighbors or communities around them, even if these neighbors or communities do not use social media [8].

### 3.2. Emotional Spatio-Temporal Trajectory Mining

Earthquakes, as a sudden disaster, cause tremendous damage in a short period of time, and is especially hazardous for of human life. Therefore, in most cities, there are many shelters for people to avoid these disasters. In this section, we explore how the spatio-temporal trajectories of human beings change when sudden disasters occur and whether these changes are related to the locations of shelters. Furthermore, in this process of change, we investigate which emotion categories are shown by human beings and how these emotion categories change. We used Chengdu as an example and determined the locations of the shelters in this city from "The Official Website of Chengdu Municipal People's Government (http://cdtf.gov.cn/chengdu/smfw/csyjbn.shtml)." Then, we translated these shelters into coordinates through the API of Baidu (http://api.map.baidu.com/lbsapi/getpoint/index.html) and vectorized the map of this region. Considering the sudden occurrence of earthquakes, we set up seven small time periods, which were 08:02 h to 08:12 h, 08:12 h to 08:22 h, 8:22 h to 08:40 h, 08:40 h to 09:30 h, 09:30 h to 10:30 h, 10:30 h to 11:30 h, and 11:30 h to 13:00 h, respectively (the earthquake occurred at 08:02 h), to analyze the changes of the crowd in these fine-grained time periods. Figure 7 shows the changes in the number of people over time. We can see that the population grew fastest in 08:40 h to 09:30 h. This may reflect that a large number of people had reached nearby shelters during this time period.

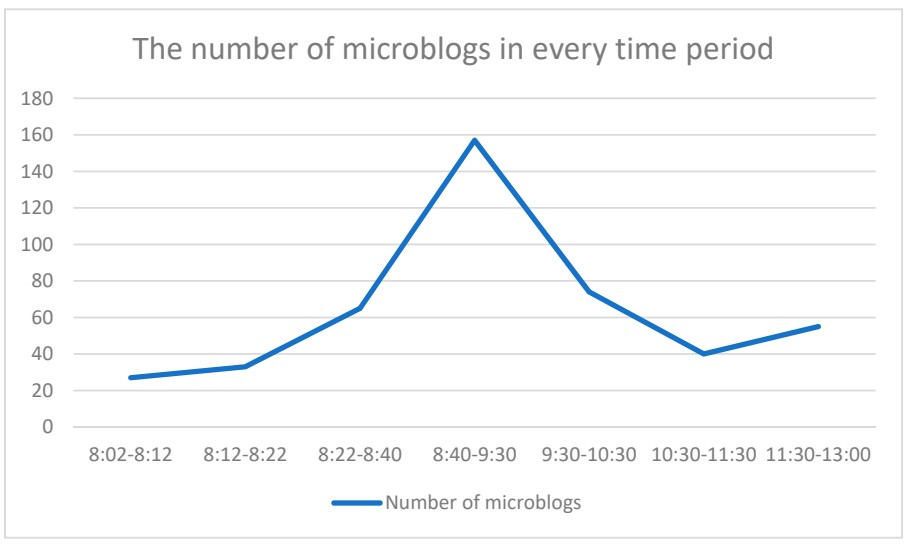

**Figure 7.** Changes of the crowd amount in each small time period.

We used the kernel density algorithm [38] to validate the change of crowd aggregation over time and explore the relationships between density centers of crowds and shelter locations, as shown in Figure 8. In Figure 8a, three clusters were formed within the first 10 minutes after the earthquake and we marked them as cluster 1, cluster 2, and cluster 3, respectively. Although the density of cluster 1 is small, we can see its core had been in the location of the shelter. This indicates that within 10 minutes of the earthquake, a small number of people had gathered in nearby shelters. Cluster 2 and cluster 3,



especially cluster 3, had higher densities than cluster 1. However, the people in these areas had not yet gathered in the shelter. Ten minutes later, between 08:12 h to 08:22 h, the number of people increased as they further converged to shelters. We found that some shelters had gathered many people, such as cluster 1 and cluster 3, as shown in Figure 8b. As time went on, a large number of discrete and small clusters were fused into a larger clusters, as shown in Figure 8c–e. This indicates that during these periods, a large number of people had reached the shelters. Among them, the number of people between 08:40 h to 09:30 h was the largest, as shown in Figure 8d. Then, the number of people gathered in shelters began to decline. But the crowd hadn't dispersed at this time, as shown in Figure 8e. We can understand these changes through the corresponding value of crowd density. In Figure 8f,g, we can see that the big cluster began to disintegrate and was decomposed of small clusters. Then, these small clusters were gradually moving away from shelters. Perhaps it means that people's emotions were no longer so tense at this time.

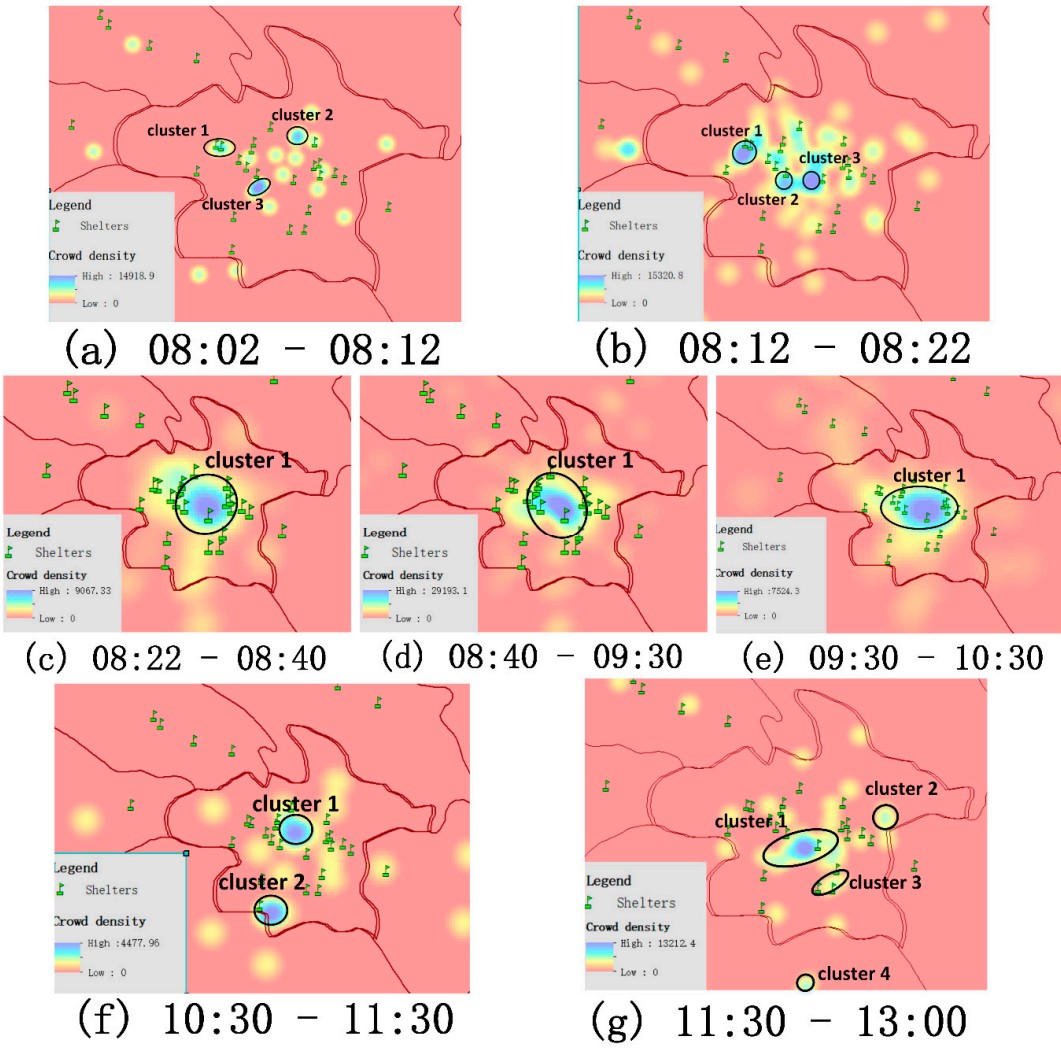

**Figure 8.** The change characteristics of public spatio-temporal trajectory. This sequence diagram describes how the crowd moved in different small time periods after the earthquake. Among them, the figure (**a**) shows the trajectories of public change in 10 minutes after the earthquake. Three clusters were formed in this period. The figure (**b**) shows the location relationship between each cluster and shelters in the second ten minutes. The figure (**c**), (**d**) and (**e**) shows that all small clusters formed a large cluster over time and it had the largest population between 08:40 and 09:00 as in figure (**d**). The figure (**f**) and (**g**) shows crowd was gradually dissipating and leaving the shelter. From the whole process of analysis, we determined that: (1) When the earthquake happened, people rushed to the

shelters in a very short time period. However, were these shelters reasonably laid out? We saw that some shelters did not contain many people, or even had no people. Therefore, the analysis results could be used as a reference for the rational layout of shelters. (2) The characteristics of crowd gathering and evacuation could be used as an effective reference to aid disaster reduction departments in dealing with future emergencies.

Further, we wanted to know which categories of emotion the public expressed during these periods and how these emotions changed because massive population movements in a short time may lead to some unnecessary accidents such as possible stampedes due to panic. Therefore, if the public emotions in this process can be monitored, it will help us take quick and effective measures to improve the efficiency of evacuation and prevent accidents. Table 1 presents the emotional characteristics in each period of time in Figure 8. Indicators in this table include clusters formed by kernel density clustering, the emotion categories and major emotion category contained in each cluster, etc.

**Table 1.** Distribution characteristics of emotions in different clusters in different time periods.

| Period of Time | Cluster | Emotion Categories | Major Emotion Category |
|---|---|---|---|
| 08:02 h to 08:12 h (Figure 8a) | Cluster 1 | Anxious | Anxious |
| | Cluster 2 | Fearful | Fearful |
| | Cluster 3 | Anxious, fearful, angry | Fearful |
| 08:12 h to 08:22 h (Figure 8b) | Cluster 1 | Anxious, fearful, neutral | Fearful |
| | Cluster 2 | Anxious, angry, fearful | Fearful |
| | Cluster 3 | Angry, fearful, positive | Fearful |
| 08:22 h to 08:40 h (Figure 8c) | Cluster 1 | Anxious, angry, Fearful, sad, neutral, positive | Fearful |
| 8:40 to 9:30 (Figure 8d) | Cluster 1 | Anxious, angry, Fearful, neutral, positive | Fearful |
| 09:30 h to 10:30 h (Figure 8e) | Cluster 1 | Anxious, fearful, Neutral, positive anxious | Anxious, fearful anxious |
| 10:30 h to 11:30 h (Figure 8f) | Cluster 1 | Fearful, positive, neutral | Fearful |
| | Cluster 2 | Neutral, sad | Neutral, sad |
| 11:30 h to 12:30 h (Figure 8g) | Cluster 1 | Fearful, neutral positive, anxious, sad | Positive |
| | Cluster 2 | Neutral, angry | Angry |
| | Cluster 3 | Angry, anxious, neutral | Anxious |
| | Cluster 4 | Neutral, positive | Neutral |

From Table 1, we can see that fearful emotions dominated in the first 150 minutes (8:02 h to 10:30 h) after the earthquake, followed by anxiousness. During this time period, people were unprepared for the unexpected earthquake and were afraid of losing their lives in the earthquake. Among them, in the first time period (8:02 h to 8:12 h), people expressed anxiousness in cluster 1, as shown in Figure 8a. Through the corresponding text content, we found that they did not know the details of the earthquake at this time (such as the location of the epicenter, the magnitude scale, etc.), so they were worried about the safety of their relatives, friends, and even others whom they did not know. Angry, neutral, and positive emotions began to appear in the second period of time (8:12 h to 8:22 h). However, the number of the positive and neutral emotions was relatively small, with one piece being positive and two pieces being neutral. Angry emotions in this time period mainly showed people's aversion to earthquakes. Sad emotions began to emerge in the third time period (08:22 h to 08:40 h) People who expressed sad emotions were mainly because this earthquake reminded them of a dreadful catastrophe

that happened in Sichuan in 2008 (the earthquake that occurred in Wenchuan, Sichuan Province, on May 12, 2008, caused great damage). With more and more detailed information about earthquakes becoming available, people expressed more categories of emotion, and the reasons for these emotions had also changed. For example, in the fourth and fifth time periods, people expressed fearful and anxious emotions because they worried about there would be some aftershocks in the near future. This also shows that for a long time, people still gathered near the shelter, as shown in Figure 8c–e. We can see many people began to leave shelters in Figure 8f. Combining with emotions people expressed in this period, we found fearful emotions were no longer dominant. People expressed sad emotions in cluster 2 in the sixth time period because of grief for the victims in the worst-hit areas. In the seventh time period, the main emotion category in each cluster was different. People might have calmed down at this time. Even the main emotion of cluster 2 in Figure 8g was positive. People expressed their prayers for the disaster areas.

The fine-grained emotion analysis was not only a further explanation for human spatio-temporal trajectory mining, but also provides more details of the disaster for disaster reduction departments. On the one hand, it provides an understanding of the public's emergency awareness and movement law in the study area. On the other hand, based on characteristics of the emotional spatio-temporal trajectory, disaster reduction departments can provide timely management and guidance for "key nodes" in this process to avoid unexpected accidents. For example, we can provide effective guidance for areas where negative emotions are intense, i.e., in Figure 8a,b, to avoid possible stampedes due to panic.

*3.3. Post-Disaster Emotional Change Analysis*

The sudden catastrophe has a long-term impact on the public. By monitoring and analyzing the public fine-grained emotional information, we can mine a lot of important information from massive disaster-related data. This information can help us quickly understand the public's needs and feedback, even for some hard-to-find problems, such as mental health. This is very important for us to improve the efficiency of disaster emergency response and rescue. In this section, we took Ya'an as an example and used days as intervals to monitor public emotions for a longer period of time from the macro perspective. Meanwhile, we analyzed the causes of public emotion change using hot word extraction. This can help us quickly grasp the public's concerns from the mass emotional information. The tool we used to extract hot words was from the web (http://www.picdata.cn/).

Regarding positive emotions, as shown in Figure 9, on the second day (April 21) after the disaster, we brought the hot words "感动 (moved)" and 感激 (grateful)" into the corresponding text. We found that the public expressed their gratitude mainly to the professional rescue workers, such as the army. There were fewer volunteers at this time. However, more and more volunteers spontaneously joined the rescue effort over time, especially on the third and fourth days. Because in this time, "志愿者(volunteer)" appear more frequently. The civilian rescue gradually arrived in the disaster area from about the fourth day (April 23) after the disaster. The hot words "爱心 (love)"and "物资 (relief supplies)" indicated that people in the disaster-stricken areas were grateful for non-governmental spontaneous relief materials. Based on the changes in public positive emotions, we can understand the general process of rescue work.

Regarding anxiety, as shown in Figure 10, as time went on, the anxiety of the public gradually decreased. On the second day after the earthquake, there was more anxiety. The reason for this was because: (1) The roads were interrupted and some areas were isolated. We used the hot words "中断 (interrupt)" and "救援 (rescue)" in the original micro-blog texts to get detailed information. We found that "上里镇 (Shangli town)," "中里镇 (Zhongli town)," "下里镇 (Xiali town)," and "碧峰峡 (Bifengxia town)" were isolated from the outside world and needed urgent rescue after the earthquake. (2) Some areas were in urgent need of relief supplies, and these areas were suffering from bad weather. For example, through hot words, we found that there were some micro-blog texts saying that "Wangjia village in Longmen town was short of water, food, medicine, and tents". (3) Some people expressed

anxiety because they could not contact their relatives and friends after the earthquake. From April 21 to April 26, we found the public's anxiety was mainly due to a lack of supplies and bad weather. Additionally, as time went on, the public's demand for relief supplies mainly concerned tents, especially on April 26th. The combination of micro-blog content and location information could have allowed a more precise rescue plan to be carried out.

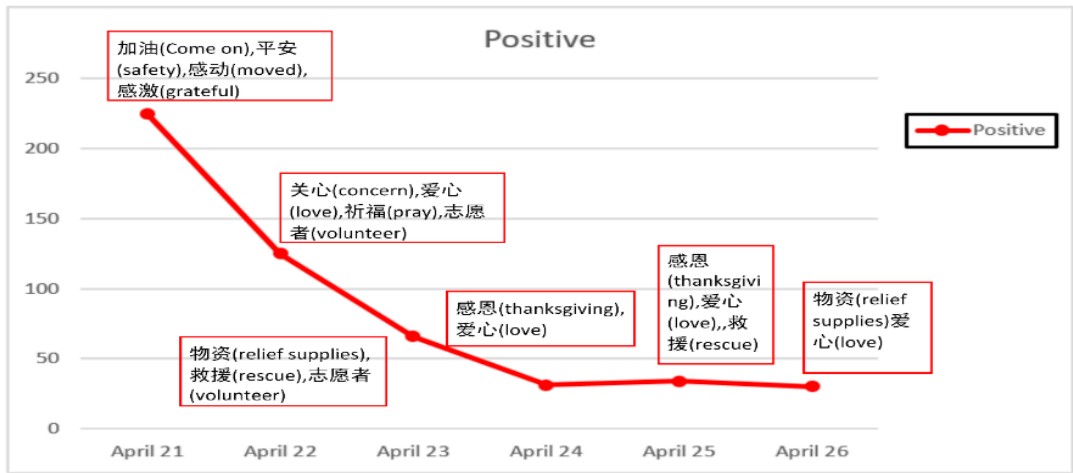

**Figure 9.** Sequence diagram of positive emotion (the words in the text box are the hot words related to this emotion in the corresponding time period).

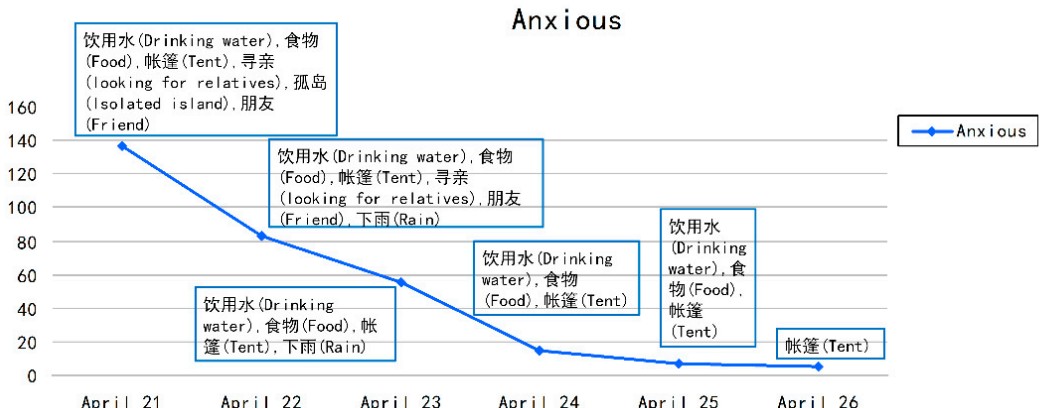

**Figure 10.** Sequence diagram of anxiety.

As shown in Figure 11, from April 21 to April 23, the public mainly expressed anger because of their aversion to the earthquake and because of some internet fraud. By combining the corresponding micro-blog content, we found that some of the Internet fraud was exposed by micro-blog users, such as "It's horrible. Some criminals cheated under the cover of the earthquake. Please pay attention to this telephone number: xxx.". These angry messages were used to help people guard against rumors (especially for anxious people). After April 23rd, public anger was mainly due to aversion against the disaster.

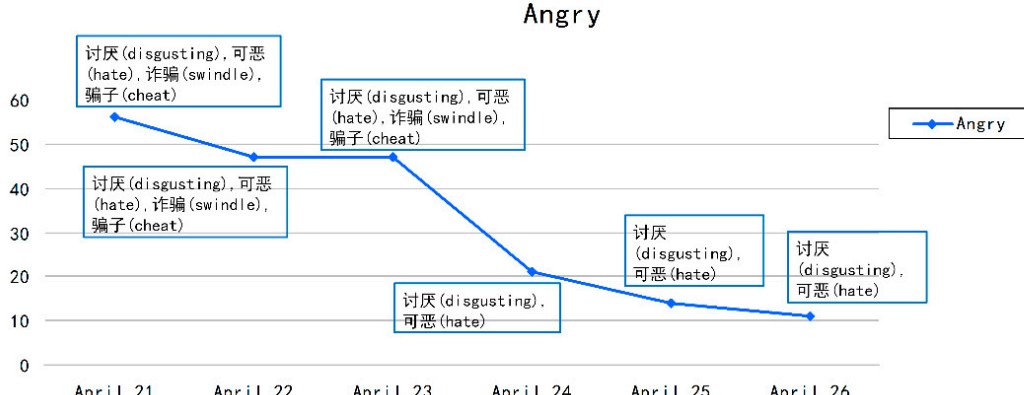

**Figure 11.** Sequence diagram of anger.

As shown in Figure 12, from April 21 to April 25, the sadness was due to the destruction of homes and the deaths of relatives or friends. Examples of corresponding micro-blog texts are: "Where is home? Where is the classroom? Yesterday? I was not an orphan yesterday" and "It's a scene of complete devastation and when can we rebuild our homeland?" On April 26, many mourning activities were carried out by official and non-governmental organizations, which was the reason why people expressed sadness. During the process of the earthquake rescue, disaster relief organizations could send psychological relief to places where sadness was intense, as determined by the locations of the corresponding micro-blog information.

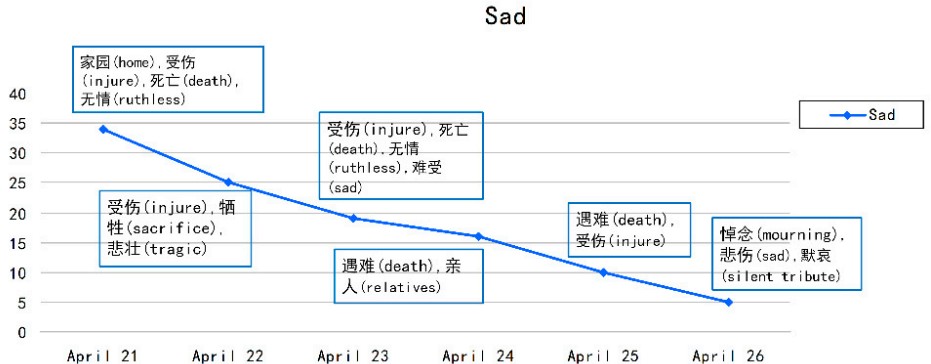

**Figure 12.** Sequence diagram of sadness.

In terms of fear, as shown in Figure 13, from April 22 to April 24, there were several aftershocks in Ya'an, which had a great impact on the public's life. Many hot words were observed, such as "余震 (aftershock)" and "可怕 (dreadful)". However, between April 24 and April 26, especially on April 26, there was a sudden increase in fear, and the hot words mainly included "心理咨询师 (psychological counselors)," "回忆 (recall)," and "惊吓 (startle)." We used these hot words to get the original micro-blog and saw some saying that: "A teacher in Zhongli Town reported that a girl was afraid of loud voices and kept eating. She said she would be afraid if she didn't eat. So this teacher hoped that the disaster reduction department could send a psychological counselor to help that girl" and "My brother said that as long as there was the thunder and lightning in Ya'an, he was very scared! Ask for help!" Therefore, the relief department should give some help to these people.

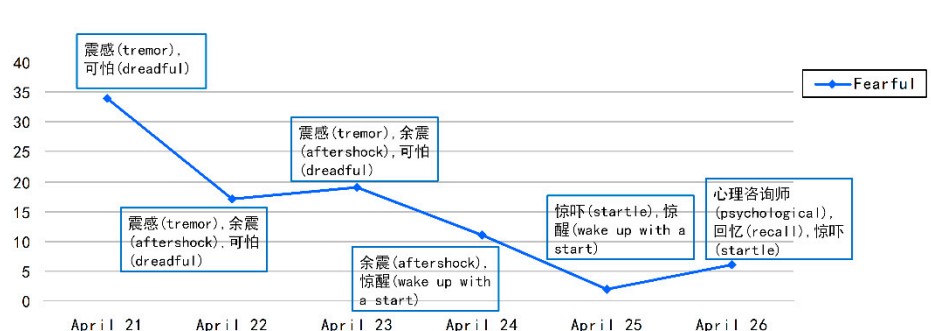

**Figure 13.** Sequence diagram of fear.

We can continue to explore other areas in the same way. This could help us to accurately understand the public's reactions to the progress of disaster mitigation. Further, the analysis results could help the rescue department to optimize rescue strategies and improve the efficiency of rescues.

## 4. Evaluation of the Experimental Indicators

### 4.1. Accuracy Evaluation of the Emotion Classification

#### 4.1.1. Experimental Corpus

In the emotion classification experiment, we first manually annotated a corpus based on the six emotion categories. In this corpus, each emotional category contained 1000 text samples. Among these, 800 text samples were selected as the training corpus and 200 were selected as the testing corpus from each emotion category.

#### 4.1.2. Experimental Environment

In order to improve the accuracy of the emotional classification, we translated special characters and symbols in the text into Chinese words. We integrated the word2vec framework (from Google [39]) and NLPIR-ICTCLAS (http://ictclas.nlpir.org) into our algorithmic framework to assist with the processing of this text. Finally, we built convolutional neural networks based on tensor flow [40], and optimized the model parameters to achieve the best results. In this process, we set the dimensions of the word vector as 200, the number of convolution kernels was 3 and their sizes were 3, 4, and 5. The size of max pooling was 4, the proportion dropout regularization was 0.3, and the stride was 1.

#### 4.1.3. Experimental Results and Accuracy Comparison

We verified the accuracy of the algorithm based on the precision (P), recall (R), and comprehensive evaluation indexes (*F-1*). The formulas are shown below:

$$P = \frac{N\_Correct}{N\_Correct + N\_False} \tag{5}$$

$$R = \frac{N\_Correct}{N\_Category} \tag{6}$$

$$F - 1 = \frac{2 \times P \times R}{P + R}. \tag{7}$$

*N_Correct* represents the number of texts that were correctly classified into one category, *N_False* represents the number of texts that were misclassified into this category, and *N_Category* represents the number of texts that belonged to this category in the testing corpus.

Table 2 and Figure 14 show the accuracy of the CNN model in the fine-grained emotion classification. The comprehensive evaluation index scores for each category were all above 81%, which met the experimental requirements. In addition, in this paper, we also considered the use of slang, internet buzzwords, and special characters and symbols to enhance the performance of the model.

**Table 2.** Accuracy evaluation of positive emotion classification.

| Emotional Category | Precision (P) | Recall (R) | Comprehensive Evaluation Index (*F-1*) |
|---|---|---|---|
| Positive | 82.25% | 80.00% | 82.54% |
| Neutral | 84.21% | 87.91% | 86.02% |
| Angry | 91.57% | 86.36% | 88.89% |
| Sad | 88.54% | 85.00% | 86.73% |
| Anxious | 78.47% | 85.27% | 81.77% |
| Fearful | 84.69% | 85.57% | 85.13% |

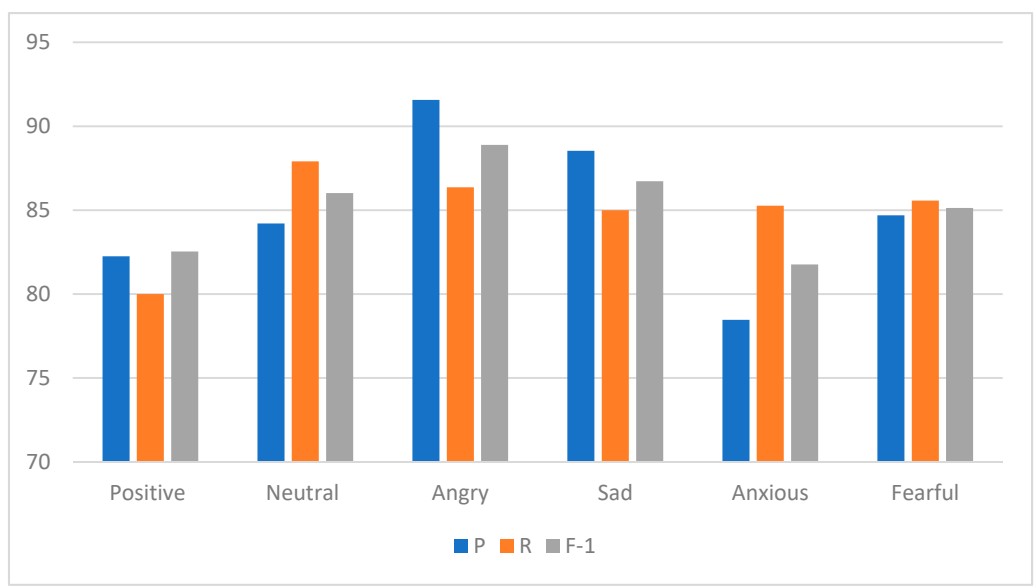

**Figure 14.** Classification accuracy of different emotions.

*4.2. Evaluation of Spatio-Temporal Analysis Experiments*

4.2.1. The Description of Data with Address Information

The number of texts in the data set of this paper was 39341. However, not all texts contained address information. We processed this address information based on the method described in Section 2.2. All address information can be divided into two categories in this paper, including rough address information and accurate address information. Among them, rough address information can only represent villages and towns, even districts and counties, such as "Lushan County, Ya'an City" and "Wuhou District, Chengdu City," etc. Accurate address information can represent streets and geographical entities, such as "Sishengci North Street," "Wangjiang Campus of Sichuan University," etc. Table 3 and Figure 15 depicts the proportion and number of data with different accuracy in different cities. Among them, the formula for calculating the proportion is as shown:

$$\text{Proportion} = \frac{\text{The number of specified texts in the city}}{\text{The number of all texts in the city}} \tag{8}$$

**Table 3.** Proportion of data with different accuracy in different cities.

| City | The Proportion of Data with Accurate Address Information | The Proportion of Data with Rough Address Information | Total |
|---|---|---|---|
| Chengdu | 8.08% | 2.16% | 10.24% |
| Ya'an | 12.18% | 13.91% | 26.09% |
| Mianyang | 9.45% | 2.60% | 12.05% |
| Leshan | 9.81% | 4.85% | 14.66% |
| Meishan | 13.30 % | 7.44% | 20.74% |
| Deyang | 8.71% | 5.79% | 14.50% |
| Aba | 11.29% | 23.51% | 34.80% |
| Ziyang | 5.10% | 3.10% | 8.20% |
| Neijiang | 4.79% | 2.33% | 7.12% |

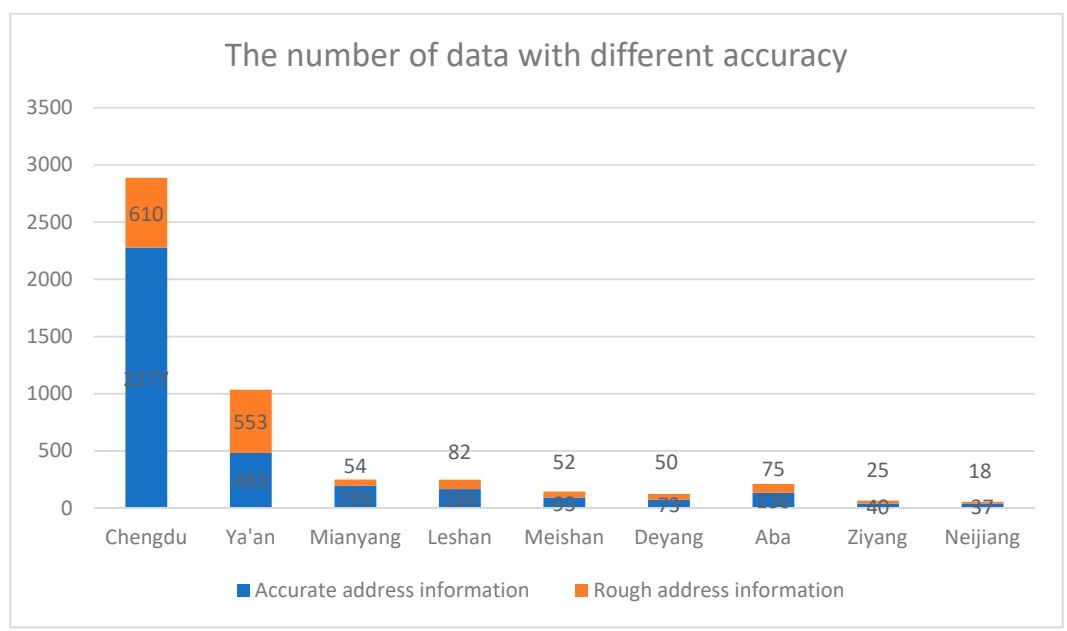

**Figure 15.** Comparison of the number of pieces of address information with different accuracy in each city.

4.2.2. Evaluation of the Experimental Process and Results

In Section 3.1, we used the population density distribution data provided by GHSL (Global Human Settlement Layer) to assist in assessing the affected population. The scale of maps we used was relatively small. Therefore, we considered that all the data with both accurate address information and rough address information can be used. Among them, the data in Aba and Ya'an better reflected the real situation expressed by social media in the region because although the social media data volume in these two cities is small, the data with address information accounted for a larger proportion; they reached 34.8% and 26.09%, respectively. Considering population density and epicenter location (the epicenter of the earthquake occurred in Ya'an), in Section 3.1, we mainly focused on the disaster situation in Ya'an and several cities close to Ya'an.

In Section 3.2, we explored how the spatio-temporal trajectories of human beings changed when sudden disasters occur and used geographic data (shelters) with accurate location information. Therefore, we considered that the social media data with accurate address information can be used for analysis. In addition, earthquakes are sudden disasters and cause tremendous damage in a short period of time. Therefore, in this paper, we set up seven small periods of time to explore the public movement after the disaster and combined this with public emotions to mine more details regarding the disaster. This required more data with accurate address information. We can see that Chengdu meets

the requirements most from Figure 15 and this produced satisfactory analysis results in Section 3.2. Of course, the same method can also be used to analyze the disaster in Ya'an; however, time granularity would be coarse.

In Section 3.3, we mainly monitored the emotional information in each city for a long time. Therefore, we just needed to know which city the social media data belongs to. Although a lot of social media data had no location information, they all had their own labels. This was explained in Section 2.1. Considering that Ya'an was the worst hit by the earthquake, we selected Ya'an as the research object.

## 5. Conclusions

When a disaster occurs, social media can provide a large amount of important disaster-related geographic information to the disaster reduction departments in near real-time. In this paper, we regarded the fine-grained public emotional information extracted from social media as an attribute of geographic information to assist in disaster mitigation. In the process of extracting emotional information, we fully analyzed the characteristics of Chinese social media and selected a suitable algorithm (convolution neural network model). Meanwhile, a large number of special characters and symbols with emotional characteristics contained in social media were also considered to improve the accuracy of classification. The methods in this paper achieved satisfactory results.

In order to verify the effectiveness of the method in this paper in disaster mitigation, we used the 7.0 earthquake that occurred on April 20, 2013, in Ya'an City, Sichuan Province, China, as a case study. We classified the social media texts related to areas affected by the earthquake into six different emotion categories. Then, with the help of GIS software and other traditional geographic information data (population density distribution data and shelter data), we explored the role of public emotional information that is helpful for disaster reduction. The results showed that fine-grained public emotions can provide more powerful data support for disaster reduction departments to optimize rescue strategies and improve rescue efficiency.

Although social media plays an important role in assisting disaster mitigation, it also has some limitations. (1) Social media data is unevenly distributed. The economically developed and populous areas tend to have more users of the Sina micro-blog. In the research area of this paper, Chengdu had the most Sina micro-blog data and these data were more concentrated in the urban area, but it was not the worst-hit city. Therefore, in future research, more abundant data that include other sources is also needed to supplement social media data, such as image data, vehicle-borne GPS data, etc. (2) Not all social media users are willing to share their location information. In the dataset used in this paper, the proportion of text with location information was very small. This limits the use of some spatio-temporal analysis methods. However, we found that there are many geographically named entities in texts and many of them can respect the user's location. Therefore, an effective method is needed to automatically extract these geographically named entities to supplement the deficiencies of geographic location information in social media.

In addition, the use of social media for disaster mitigation is far from enough. With the development of data mining technology, more disaster-related information contained in social media can be extracted, such as different categories of disaster loss information, etc. With the help of the powerful spatio-temporal analysis ability of GISs, this useful information can play a greater role.

**Author Contributions:** T.Y., J.X. and G.L. conceived and designed the paper; T.Y. and J.X. wrote the paper; T.Y. and N.M. designed and implemented the algorithmic framework; T.Y. and Z.L. realized the visualization; C.T. and J.Z. collected data and processed them.

**Funding:** This research was funded by the National Key R&D Program of China, grant number 2016YFE0122600, the National Natural Science Foundation of China, grant number 41771476 and Strategic Priority Research Program of Chinese Academy of Sciences, grant number XDA19020201.

**Acknowledgments:** We thank Edward T.-H. Chu, Associate Professor at Yunlin University of Science and Technology for his advice and collaborative work on the emotion classification under the framework of cooperation project. We also thank Zhenyu Lin and Qinglan Zhang, Master students at Henan Polytechnic University for their advice on visualization.

**Conflicts of Interest:** The authors declare no conflict of interest."  must be declared in this section. If there is no role, please state "The funders had no role in the design of the study; in the collection, analyses, or interpretation of data; in the writing of the manuscript, and in the decision to publish the results".

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
