# Peer review of "Social Media Big Data Mining and Spatio-Temporal Analysis on Public Emotions for Disaster Mitigation"

_ijgi, doi:10.3390/ijgi8010029_

Round 1
Reviewer 1 Report
My main reason for recommending the rejection is that the paper is out of the scope of the IJGI journal. The two main ideas of the paper when one reads the abstract is the use of CNN (Convolutional neural networks) alongside emotional information analysis and classification. The only reference to spatial analysis is the last sentence of the abstract, which is clearly secondary to the main objectives of the paper.
As expected, the rest of the paper reports on the use of CNN to analyse, process and classify short messages from social media services into emotions. Only in section 4 (on page 10!), there is a reference to "spatio-temporal analysis". Unfortunately, the authors limit it to show points on a map.
Authors: please, do not get me wrong. I am fully convinced the paper has merit and could be published in outlets related to text mining and data mining fields, for example. In my opinion, the paper does not provide scientific contribution in the GI science field, therefore I do not believe it can be published in a GI Science related journal like IJGI.
Author Response
Thank you very much for your advice on my paper.
I am sorry that we have not been able to clearly express the topic of this paper for you. Social media was a component of crowdsourcing geographic data. It plays an important role in disaster mitigation. At present, many geoscientists have studied how to use social media to serve disasters mitigation. But related research focuses more on the spatio-temporal information in social media and ignored disaster-related information. In this paper, we regarded the public emotional information contained in social media as attribution data of geographic information and consider that these emotional information can reflect more details of the disaster. But it's a great pity that there is not much spatio-temporal analysis method in this paper. So we seriously revised this paper, even adjusted its structure to make the topic of this paper more reasonable. In the original discussion section, we added a lot of spatio-temporal analysis to make up the shortcomings in this respect. We combined population density distribution data to study the changes in population-affected areas. Combining the public emotions, it can help to ensure effective assessment of the disaster situation and rational deployment of rescue resources. It is generally believed that negative emotions indicate that the earthquake has had a greater impact on the people. We also explore crowd trajectory law after the earthquake with the help of GIS powerful analysis function. Emotions are used to express public feedback in the mobile process. This is important for emergency response by disaster reduction department. But meanwhile, it is necessary to introduce the emotion extraction algorithm in detail. This can help readers understand the extraction process and precision to ensure the rationality of spatio-temperal analysis in this paper.
Thank you again for your comments. We have made major revisions to the paper and hope these changes can better express our topic.
Reviewer 2 Report
Thank you very much, this is a very interesting paper and spatio-temporal analysis of social media data.
Comments:
- English editing is required (especially first tow sentences of abstract)
- Figure 2 (Map) should be improved. The text is hard to read, can you maybe increase the size, and/or use a brighter color, or a white buffer around text, or something to improve the cartography? Also an overview where exactly in China this Map is, would be good for orientation for readers who do not know much about China geography.
- All formulas are in bad graphic quality. Please use the formula editor if you write in Word, or use the LaTex formula notation if you write in LaTex, to guarantee vector symbols in the formulas.
- I am not sure if political suggestions, should be part of a scientific paper. For example, in l. 443 - 445 you write, that the government should increase screening of network information in context to "fake information". At least where I am from (Europe), a suggestion like this would be regarded quite critical. Because screening of network information leads to censorship and surveillance and is (most of th e time) followed by oppression of the people by the government.
Author Response
Thank you very much for your pertinent comments on my paper.
Point 1: English editing is required (especially first tow sentences of abstract)
Response 1: We have made a reasonable adjustment to the structure and improve the language in this paper.
Point 2: Figure 2 (Map) should be improved. The text is hard to read, can you maybe increase the size, and/or use a brighter color, or a white buffer around text, or something to improve the cartography? Also an overview where exactly in China this Map is, would be good for orientation for readers who do not know much about China geography.
Response 2: The Fig 1 and Fig 2 have been modified to make the presentation clearer.
Point 3: All formulas are in bad graphic quality. Please use the formula editor if you write in Word, or use the LaTex formula notation if you write in LaTex, to guarantee vector symbols in the formulas.
Response 3: All formulas in this paper have also been modified as required.
Point 4: I am not sure if political suggestions, should be part of a scientific paper. For example, in l. 443 - 445 you write, that the government should increase screening of network information in context to "fake information". At least where I am from (Europe), a suggestion like this would be regarded quite critical. Because screening of network information leads to censorship and surveillance and is (most of the time) followed by oppression of the people by the government.
Response 4: I think that's a good suggestion. And I have revised and adjusted the relevant content in the paper.
Thank you again for your valuable suggestions.
Reviewer 3 Report
This paper is very interesting and well presented.
I have a few minor comments:
- Introduction section: Better introduce the emotional aspect (which is central for the presented study). It is only briefly introduced on lines 74-76. It would be good to strengthen this aspect to increase readers understanding.
- Section 2.1: I was wondering whether the Sina micro blog has an API? If yes why authors have not used it to gather data?
- That would be good to have some perspectives in the discussion section to understand how the proposed approach can be effectively used/implemented in disaster relief?
Author Response
Thank you very much for your pertinent comments on my paper.
Point 1: Introduction section: Better introduce the emotional aspect (which is central for the presented study). It is only briefly introduced on lines 74-76. It would be good to strengthen this aspect to increase readers understanding.
Response 1: I have adjusted the structure of the paper to make it looked more reasonable. And I introduce the role of public sentiment in disaster reduction in more detail in this paper.
Point 1: Section 2.1: I was wondering whether the Sina micro blog has an API? If yes why authors have not used it to gather data?
Response 2: Sina Weibo platform provides API interface to get their data. But at present, these interfaces provide very limited functionality and can not meet the needs in my study. We explained it in the original text “Social media platforms usually provide an interface or API that allows developers to retrieve social media data. However, the retrieval of data in this way has great limitations, for example, you cannot set the time-span and topics, etc. So, in this paper, we used Sina micro-blog's advanced search capability to get data by using time-span, location, and event-related key words.”
Point 3: That would be good to have some perspectives in the discussion section to understand how the proposed approach can be effectively used/implemented in disaster relief?
Response 3:In the discussion section,we revised the original content and added spatio-temporal analysis. Meanwhile, more viewpoints and discussions were introduced.
Thank you again for your valuable suggestions.
Reviewer 4 Report
The research work under review tries to determine the emotional states of people before a catastrophe from the information published in the social media in order to give support to the authorities in decision making.
Despite having made an adequate introduction of the problem, the authors do not make clear what research questions are posed or what are the starting hypotheses. They propose to directly analyze the use of CNNs to classify emotional information and compare the results with other classification techniques used in NLP such as Naive Bayes, K-Nearest Neighbor or Support Vector Machine, and analyze the results obtained from a temporal space perspective.
The authors have not made a good state of the art regarding the NLP with Neural Networks, there have been studies for several years with different types of networks such as Recurrent Neural Networks (RNN) or Hierachical Attention Networks (HAN).
Nor do they justify later why in the analysis of the 2003 earthquake data, 7 days are studied and the study is carried out with a temporary granularity of one day. All the feelings and problems in the face of a catastrophe such as an earthquake are unleashed and happen in short periods of time. The 48 first hours could have been analyzed with granularity of one hour.
The authors also have not justified the use of six emotional states: positive, neutral, angry, anxious, fearful and sad. Based on what criteria, study, references do you decide to use these 6 states?
As for the spatial analysis carried out, indicate that it is only qualitative. No type of analytics is done on the distribution of the data. It is true that it is extremely complicated to assign a location to this type of information messages in social media. But this does not prevent analysis from being carried out with the few data that are georeferenced.
All these facts lead me to propose that the work be rejected and that the authors can redo the research work strengthening the indicated weaknesses: state of the art, justifications based on own or other authors' studies, analysis of the data with granularity of time, non-quantitative spatial study.
Also indicate to the authors that they must take care in order to not change the terms used in the manuscript (spatio-temporal, spatiotemporal), that they review the small mistakes/misspelling in the text, that clarify better the figure 1 regarding the flow that joins Test text with Making text feature vector (In the Emotional classification part), do not use can’t in scientific text and you must reference the equations by their number among other issues.
Author Response
Thank you very much for your pertinent comments on my paper.
Point 1: Despite having made an adequate introduction of the problem, the authors do not make clear what research questions are posed or what are the starting hypotheses. They propose to directly analyze the use of CNNs to classify emotional information and compare the results with other classification techniques used in NLP such as Naive Bayes, K-Nearest Neighbor or Support Vector Machine, and analyze the results obtained from a temporal space perspective.
And
Point 2: The authors have not made a good state of the art regarding the NLP with Neural Networks, there have been studies for several years with different types of networks such as Recurrent Neural Networks (RNN) or Hierachical Attention Networks (HAN).
Response 1 and 2: According to your suggestion, we have made a reasonable adjustment to the structure in this paper. In the introduction, we adjusted the way the topic was expressed. We systematically enumerate the application of public emotion in disaster mitigation. The CNN method was used to extract public emotions from texts in this paper. It is a part of the framework for disaster mitigation and used to assist analysis. As your advices, there are many algorithms to extract public emotions, such as HAN, RNN, etc. Before we started this work, we also read many literatures to understand these methods. Combining the characteristics and usage scenarios of these methods, we finally chose CNN. In order to make the reader more clear, we added the content why we selected CNN in this paper. Meanwhile, in the experiment section, we deleted the comparison with traditional machine learning methods.
Point 3: Nor do they justify later why in the analysis of the 2003 earthquake data, 7 days are studied and the study is carried out with a temporary granularity of one day. All the feelings and problems in the face of a catastrophe such as an earthquake are unleashed and happen in short periods of time. The 48 first hours could have been analyzed with granularity of one hour.
Response 3: Earthquake is a sudden natural disaster and can cause tremendous damage in a short period of time. We very much agree with your proposal. So we added fine-grained analysis in this paper. We use the spatio-temperal analysis method of GIS to explore crowd trajectory law after the earthquake. Time granularity set to minutes. Emotions are used to express public feedback in the mobile process. We also combined population density distribution data to study the changes in population-affected areas according to public emotions. It is generally believed that negative emotions indicate that the earthquake has had a greater impact on the people. Time granularity is based on hours. We divide the time periods according to “rescue-72 hours”. For monitoring of long time series, we consider that it is also an aid to disaster mitigation. On one hand, this can provide us with a reference from a macro point of view, such as what are the public's concerns, how public emotion changes over time and what is the effect of disaster relief(This can be reflected by changes in public emotion. It may take longer time for some damage to recover.) On the other hand, some of the damage may take a long time to show up, such as mental health problems. Of course, this may take longer time. But social media has a limited life cycle on events. In this earthquake, we found that seven days later, there was very little news about it. But the ideas may still be useful.
Point 4: The authors also have not justified the use of six emotional states: positive, neutral, angry, anxious, fearful and sad. Based on what criteria, study, references do you decide to use these 6 states?
Response 4: In this paper, we introduced six emotions. I'm sorry I didn't specify them clearly. I added the related content to the paper. I enumerate some bases why we divided them in to 6 kinds of different emotions in this paper.
Point 5: All these facts lead me to propose that the work be rejected and that the authors can redo the research work strengthening the indicated weaknesses: state of the art, justifications based on own or other authors' studies, analysis of the data with granularity of time, non-quantitative spatial study.
Response 5: Thank you for your advice. We have added other geographic data and with the help of spatio-temporal analysis method of GIS to analyze these data. The public emotions contained in social media can be regarded as attribute data of geographic information to assist spatio-temporal analysis. In future studies, we will further highlight the research of spatio-temporal analysis methods.
Point 6: Also indicate to the authors that they must take care in order to not change the terms used in the manuscript (spatio-temporal, spatiotemporal), that they review the small mistakes/misspelling in the text, that clarify better the figure 1 regarding the flow that joins Test text with Making text feature vector (In the Emotional classification part), do not use can’t in scientific text and you must reference the equations by their number among other issues.
Response 6: Thank you for reminding us of the details in this paper. We have amended the relevant parts.
Thank you again for your valuable suggestions.
Round 2
Reviewer 1 Report
Dear authors,
I was surprised to see the radical change that the newspaper suffered. The new structure and narrative of the article is more convincing. The authors also provide additional paragraphs to justify the decision made in terms of classification methods and analytical methods used throughout the paper. Figure 1 now drives the structure of the paper, especially for section 2. An improvement may be to indicate the corresponding subsection for each box in the figure.
As regards the author's response, I would say that social media is **not** a component of the crowdsourced geographic data . Location and geographic information are important, but crowdsourced geographic data , as we know them today, are derived from social network services and platforms. Without the latter, the first has not yet existed. It is clear that the authors paid more attention to spatial analysis. The novelty is not in the methods used, but in the interpretation of the data through the geospatial footprints. I'm still not really convinced of section 3, but I would say it has improved a lot. Figure 6 and 8 are vital for giving a geospatial interpretation of the analysis. However, I cannot read them. It is a pity that the authors have spent so much in re-writing the paper and these figures are basically usefulness and unreadable to the average human eye. Place, improve them and give the importance they deserve.
On a final note, conclusion is not only limited to a summary of the paper. I would expect something else from the authors . Any lesson learnt? How can the result be interpret by different stakeholders? etc.
@authors: good job!
Author Response
Question1: As regards the author's response, I would say that social media is **not** a component of the crowdsourced geographic data . Location and geographic information are important, but crowdsourced geographic data , as we know them today, are derived from social network services and platforms. Without the latter, the first has not yet existed. It is clear that the authors paid more attention to spatial analysis. The novelty is not in the methods used, but in the interpretation of the data through the geospatial footprints. I'm still not really convinced of section 3, but I would say it has improved a lot. Figure 6 and 8 are vital for giving a geospatial interpretation of the analysis. However, I cannot read them. It is a pity that the authors have spent so much in re-writing the paper and these figures are basically usefulness and unreadable to the average human eye. Place, improve them and give the importance they deserve.
Thank you very much for your correction. I had revised the corresponding content. In addition, in order to increase the credibility of the analysis, I described the acquisition process of data location information and accuracy of data in all spatio-temporal analysis. It is generally known that social media data comes from the “unconscious behavior” of the public. It may be more complex than crowdsourcing data. So the shortcomings of data missing and incomplete are relatively serious. But its advantages for disaster reduction are also outstanding (real-time, low-cost and huge amount of data, ect). In section 4 in this paper, I supplemented the accuracy of the data to clarify the pros and cons of social media data and analysis case in this paper. I am very sorry that the Figure 6 and 8 I made was confusing to you. So I reproduced them and gave the necessary instructions.
Question2: On a final note, conclusion is not only limited to a summary of the paper. I would expect something else from the authors . Any lesson learnt? How can the result be interpret by different stakeholders? etc.
That's a great suggestion. I revised the conclusion in this paper. I analyzed and summarized the lessons in using social media data and gave corresponding advises. On one hand, the distribution of social media data is uneven. Data are abundant in developed areas. In this paper, Chengdu, as the capital of Sichuan Province, has the most data and these data are mainly concentrated in urban areas. So different data sources should be considered to overcome this shortcoming, such as GPS data and even Wechat data, etc. On the other hand, the number of data containing location information in this paper is small. To a certain extent, this limits the application of many spatio-temporal analysis methods. But we find that social media texts contain a large number of geographically named entities, which can represent the location information of users. So an automated method of identifying and extracting these geographically named entities is necessary. It can improve the efficiency of social media use. In addition, social media also contains other disaster-related information, such as disaster loss information. This needs us to continue to explore.
Thank you again for your valuable suggestions. These suggestions are very helpful for my research.
Reviewer 4 Report
The authors have improved the manuscript, have better adjusted the title of the contribution and as a whole has improved markedly.
The proposed issue of clearly defining the research question relating to the challenge has not been addressed.
I maintain that being more specific when raising the question of research helps the reader to understand the article and the methodological proposal.
Although I did not raise it directly in the first review, the spatial component used (spatiotemporal) is not very much a contribution. This type of analysis is common and basic in the field of GIS. It is not indicated how is the location obtained from blogs. Is it textual - an address? Are they coordinates? How are these data obtained? How precise are they?
Just as a precision analysis of the results has been included in section 4, an analysis of the spatial component would be expected. In which province can the Location component be more precise?
In addition to these issues of greater relevance, there are still errors of lack of space between an acronym and its definition included in parentheses (line 69), some point that remains ... (line 78). Please check it.
In figure 14, the legend informs of F, although comprehensive evaluation index is (F-1) (line 565). Is it okay F or is F-1?
Finally indicate that the text was expected to be clean, without the control of changes, but highlighting in some way the changes made in the document. It is difficult or impossible to determine which part of it has been changed.
Author Response
Question1: The proposed issue of clearly defining the research question relating to the challenge has not been addressed.I maintain that being more specific when raising the question of research helps the reader to understand the article and the methodological proposal.
I'm sorry I didn't make it clear at the beginning of the paper. I have revised the introduction again, highlighting the problems and solutions of this study.
Question2: Although I did not raise it directly in the first review, the spatial component used (spatiotemporal) is not very much a contribution. This type of analysis is common and basic in the field of GIS. It is not indicated how is the location obtained from blogs. Is it textual - an address? Are they coordinates? How are these data obtained? How precise are they?
That's a great suggestion. I did ignore it. I supplement and introduce them in Section 2.1 of this paper. Location information in social media is address. The accuracy of these addresses is also different. I introduced how to partition these different precision addresses and convert them into coordinates.
Question3: Just as a precision analysis of the results has been included in section 4, an analysis of the spatial component would be expected. In which province can the Location component be more precise?
That's still a good suggestion. I think it is necessary to evaluate the effectiveness of the spatio-temporal analysis. I supplement them in Section 4 of this paper. First, I give an overview of the data with different location accuracy information. Then the application of these data in spatio-temporal analysis is discussed. Further I also explore the analytical accuracy in different cities. This is related to the number of the data with different location accuracy information. In addition, I have revised the figure in Section 3 to increase its expressiveness. Among them, in Section 3.2, I adjusted the time granularity to achieve better results.
Question4: In addition to these issues of greater relevance, there are still errors of lack of space between an acronym and its definition included in parentheses (line 69), some point that remains ... (line 78). Please check it.
In figure 14, the legend informs of F, although comprehensive evaluation index is (F-1) (line 565). Is it okay F or is F-1?
Thank you for discovering these minor mistakes. I have corrected them in the paper.
Question5: Finally indicate that the text was expected to be clean, without the control of changes, but highlighting in some way the changes made in the document. It is difficult or impossible to determine which part of it has been changed.
I'm sorry I didn't use the “Revisions Mode”. Because previous modifications were relatively large. In this revision, we used the “Revisions Mode”.
Thank you again for your valuable suggestions. These suggestions are very helpful for my research.
Round 3
Reviewer 4 Report
Thanks to the authors for the effort devoted to improving the article and attending to the suggestions of the previous reviews.